# Primary Arthroplasty or Internal Fixation in Intertrochanteric Femur Fractures: A Survey of Surgical Attitudes of Orthopedic Surgeons in Turkey

**DOI:** 10.3390/geriatrics7010018

**Published:** 2022-02-14

**Authors:** Hakan Cici, Yunus Emre Bektas, Nihat Demirhan Demirkiran, Ramadan Ozmanevra

**Affiliations:** 1Department of Orthopedics and Traumatology, Izmir Katip Celebi University, Ataturk Education and Training Hospital, Karabaglar, Izmir 35100, Turkey; ortocici@outlook.com; 2Department of Orthopedics and Traumatology, Gaziemir Nevvar Salih Isgoren State Hospital, Gaziemir, Izmir 35100, Turkey; dr.emrebektas@gmail.com; 3Department of Orthopedics and Traumatology, Kutahya Health Sciences University, Evliya Celebi Education and Research Hospital KUTAHYA, Kutahya 43060, Turkey; drdemirhandemirkiran@gmail.com; 4Department of Orthopedics and Traumatology, University of Kyrenia, Kyrenia 99300, Cyprus

**Keywords:** arthroplasty, intertrochanteric, femur, fracture, survey

## Abstract

This study aimed to examine the primary arthroplasty attitudes of Turkish orthopedics and traumatology specialists and residents to patients with intertrochanteric fractures, of various ages, comorbidity situations and fracture types, using an internet-based questionnaire. Between March and April 2021, a cross-sectional survey was conducted with a total of 159 orthopedics and traumatology specialists and residents in Turkey, using an online questionnaire that consisted of 16 different patient scenarios of varying intertrochanteric fracture types, ages, and comorbidity conditions. Respondents’ preference ratio for primary arthroplasty was 24.1% in the scenarios with patients over the age of 71, while it was 8.4% in the scenarios with patients aged between 50 and 70. The ratios of primary arthroplasty preference were 12.4%, 21% and 27.3% in 2-part, 3-part and 4-part fracture scenarios, respectively. The primary arthroplasty preferences of respondents with 10 years or more of professional experience were observed to be statistically significantly different to those of the respondents with 1 to 10 years of experience in the 4-part fracture scenario where the patient was aged 71 years and above with an ASA (American Society of Anesthesiologists) score of 3–4 (*p* < 0.05). Despite varying opinions in the literature in recent years, primary arthroplasty can be considered a valuable alternative approach for Turkish surgeons, and in older adult patients with unstable intertrochanteric fractures, particularly those who need early mobilization and have high ASA scores.

## 1. Introduction

Intertrochanteric femoral fractures (IFF) account for approximately half of all hip fractures in older adult populations, and early treatment is of great importance to prevent perioperative and long-term mortality and morbidity associated with these injuries [1,2]. The currently accepted general treatment approach is a stable fixation and immediate weightbearing mobilization [3]. However, primary arthroplasty has been suggested as an alternative to internal fixation techniques, particularly in selected patient groups with unstable and comminuted fracture patterns, based on the experiences obtained from internal fixation methods over the decades [4,5]. Factors such as a loss of reduction, nonunion, and implant failure following internal fixation in older adult subjects with poor bone quality pose a vexing problem for surgeons. Additionally, internal fixation methods may not always meet the need of full weightbearing mobilization protocols in the early postoperative phase in patients with high comorbidities [6]. Furthermore, it has been previously shown that, in addition to factors specific to the patient and fracture type, surgeons’ own professional experience can be a significant factor in the management of IFF [7].

The aim of this survey study is to examine the surgical method preferences of Turkish orthopedics and traumatology specialists and residents in patients with IFF of various ages, comorbidity situations and fracture types, using an internet-based questionnaire, and to assess the findings in light of current literature. We hypothesize that arthroplasty is the mostly preferred approach in Turkey for orthopedic surgeons for treating older adult patients with intertrochanteric femur fractures.

## 2. Materials and Methods

The study involved 159 participants who voluntarily agreed to engage in the survey and; participants were currently employed as orthopedics and traumatology specialists or residents at university hospitals, training and research hospitals, state hospitals, and private health institutions in Turkey, between March and April 2021. The survey participation rate was approximately 10% (159/1553). The questionnaire, which consisted of 16 multiple-choice questions and was created on the internet using a free server (www.docs.google.com, accessed on 13 May 2021), was distributed to participants by uploading the server connection to various professional social networking platforms, where only orthopedics and traumatology specialists and residents were present. A restriction on the Internet protocol (IP) was used to prohibit participants from participating more than once.

In the questionnaire, the age, current practice setting, the institution where they received their residency training (university hospital, training and research hospital), and their number of years in practice (1–10 years, >10 years) were questioned in order to determine the demographic characteristics of the respondents. Afterwards, 16 questions were obtained in total, by constructing four patient scenarios for each of four demonstrative hip X-rays selected representing 2-part, 3-part, 4-part and reverse-oblique IFF; they differed from each other in terms of age and the ASA (American Society of Anesthesiologists) physical status score for each (Table 1). In the scenarios, the term of 71 years and over was used to define advanced-age patients [8]. The parameters used to characterize patient comorbidity were: an ASA score of 1–2 for a healthy or mild-systemic-disease patient, and an ASA score of 3–4 for a patient with severe systemic disease that limits activity or presents a permanent life-threatening condition [9]. The responses were recorded for 16 patient scenarios of varying fracture types, ages, and comorbidity conditions, enabling participants to pick any one of the treatment options, which included internal fixation methods (proximal femoral nail, dynamic hip screw, plate, etc.) or primary arthroplasty.

SPSS (Statistical Package for the Social Sciences 20.0, IBM Corp., Chicago, IL, USA) was used in the analysis of the obtained data. Continuous data were expressed as mean and standard deviation, and categorical data were expressed as frequency and percentage. Fisher’s exact test was used when the chi-square test requirements were not met in the comparison of categorical data. The statistical significance level was accepted as *p* < 0.05. Cramer’s V was used for the measurement of effect size.

## 3. Results

Table 2 shows the demographics of the respondents who completed the questionnaire. The average age of the respondents was 38.2 (SD:7.9). Thirty-two percent (*n*:51) of the respondents were working in training and research hospitals. While sixty-nine percent (*n*:109) of them completed their residency training at university hospitals. Sixty-four percent (*n*:101) of the respondents were found to have 1 to10 years of professional experience.

Respondents working in academic settings, such as university or training and research hospitals, were found to have a higher rate of internal fixation preference than respondents working in other hospitals in all patient scenarios, but the difference was not statistically significant (*p* > 0.05) (Table 3).

The findings regarding the surgical treatment method preferences of the respondents based on their experience (number of years in practice) are demonstrated in Table 4. Although physicians with 10 years or more of professional experience were more likely to have arthroplasty preferences than physicians with 1 to 10 years of experience in all scenarios, the difference was statistically significant, particularly in scenarios of patients aged 71 or above who had an ASA score of 1–2, in the 2-part, 3-part, or 4-part intertrochanteric fracture scenarios (*p* < 0.05). Moreover, the arthroplasty preferences of respondents with 10 years or more of professional experience were observed to be statistically significantly different to the respondents with 1 to 10 years of experience for 50–70 year-olds with ASA scores of 3–4 in the 3-part fracture scenario; and for patients aged 71 years and above, with ASA scores of 3–4 in the 4-part fracture scenario (*p* < 0.05). For patients aged 71 years and over with ASA scores of 3–4 in the 4-part fracture scenario, primary arthroplasty preference rates were 40.6% and 59.6% of respondents, with 1 to 10 years and more than 10 years of experience, respectively. No statistically significant difference was observed in surgical attitudes between respondents who had a University-Hospital-based training background and a Training-and-Research-Hospital-based training background (*p* > 0.05).

Figure 1 depicts the treatment method preference rates based on patients’ age (a), IFF type (b), and ASA scores (c). It was observed that the respondents’ preference for internal fixation is likely to decrease in the scenarios of patients over the age of 71 with advanced fracture instability and higher ASA scores. The rate of arthroplasty selection was found to be 4.2% in all patient scenarios of the reverse oblique fracture type.

## 4. Discussion

The basic treatment principle of intertrochanteric femoral fractures (IFF) consists of stable fixation and immediate weightbearing mobilization in the early postoperative period. While internal fixation techniques may easily treat stable fractures, risk factors such as low bone quality and multi-part unstable fracture patterns in older adult patients indicate that primary arthroplasty could be an alternative treatment option [10]. Although a limited number of studies have suggested that older adult patients with osteoporosis should be treated with primary arthroplasty to avoid the possible risk of secondary reduction loss, these studies have not received widespread support in the literature [11,12]. However, given the limited recommendation in the literature regarding the primary arthroplasty option, there is not enough information about the extent of arthroplasty attitudes of orthopedic surgeons in their own clinical practice. To the best of our knowledge, this is the first study in the literature investigating the arthroplasty attitudes of orthopedic surgeons in IFF.

Treatment attitudes toward IFF were examined in a survey conducted in 2013 with the participation of 239 German orthopedic specialists, but primary arthroplasty was not provided as a treatment choice to the participants in the study. Although the participants’ additional treatment recommendations were queried in a separate field using the open-ended method, no response to arthroplasty was obtained in the study. We think that the questions asked independently of factors such as age, activity status, and accompanying comorbidity, in addition to the type of fracture, may have had an effect on the participants’ preferences [13]. In our study, we observed a tendency toward primary arthroplasty with the increasing complexity of the fracture. We found it remarkable that in the scenario of the patient over the age of 71 with an ASA score of 3–4 and an unstable 4-part fracture, the surgeons’ preferences toward primary arthroplasty was approximately half. Except for the scenario of the patient over 71 years old with an ASA score of 3–4, we observed that the respondents’ attitudes toward arthroplasty were reasonably low for reverse oblique intertrochanteric fractures. The accepted treatment principle for reverse oblique fractures in the literature appears to be stable fixation, except for patients who need early mobilization, as well as for those with short remaining lifespans, which is consistent with our current findings [14].

Poor bone quality and fracture instability constitute the common basis of studies advocating for primary arthroplasty for the treatment of IFF in older adults. On the contrary, in young patients treated with internal fixation, low complication rates (avascular necrosis and nonunion rates of less than 1%) left no room for debate in this field [15,16]. In our survey study, almost all of the respondents tended to prefer internal fixation techniques for the patient aged 50 to 70 years with an ASA score of 1–2 in the 2-part fracture scenario. However, in the same age group, higher ASA scores and advanced fracture instability seemed to have an effect on surgeons’ arthroplasty attitudes. In fact, there are rare indications of this in young patients, such as accompanying ipsilateral inflammatory hip joint disorder, osteoarthritis, malunion, nonunion, and pathological fractures for IFF [17].

Several studies have compared primary arthroplasty with internal fixation in older adults with IFF. However, no consensus has yet been reached on the ideal surgical approach [18]. While longer operation times were recorded in primary arthroplasty, it is thought that surgical experience is more significant in primary arthroplasties conducted on the basis of IFF, and that the heterogeneity of surgeons with different levels of experience is what led to the findings in the studies [19]. This condition may also explain why respondents with 10 years or more of clinical experience who took part in our questionnaire study were more likely to opt for primary arthroplasty.

It was stated that, in terms of hospital stay, there was no significant difference between the two techniques, while internal fixation techniques resulted in less blood loss and required fewer transfusions, as one would expect given the shorter surgical period and applicability of minimally invasive techniques [14,20,21,22].

In long-term follow-up, internal fixation techniques have been stated to provide better outcomes [14]. However, there is a broad consensus that patients treated with arthroplasty have improved functional outcomes, particularly in the first six months following surgery, because it allows for early full-weightbearing mobilization [22]. Factors such as varus collapse, medialization (loss of offset), and shortening due to excessive sliding—possibly occurring as a result of the fixation of unstable IFFs with implants such as dynamic hip screws and proximal femoral nails—may trigger gait issues by affecting hip mechanisms. While partial weightbearing mobilization protocols are advised to minimize the risk of fracture collapse in the early postoperative period, a lack of cooperation in older adults makes these protocols difficult to follow. Furthermore, it has been considered that neglecting to reconstruct the trochanter major and abductor mechanism during minimally invasive internal fixation techniques may cause postoperative limping [6,23,24]. The reconstruction of the abductor mechanism is essential in IFF treated with primary arthroplasty. In cases in which the trochanteric bone stock and size are adequate, cerclage wires and cable-fixation techniques are useful. Although hook plates are an alternative in similar cases, complications such as soft-tissue irritation, bursitis, and nonunion are common. While nonunion in the fracture line is common after abductor mechanism reconstruction, it is usually asymptomatic and tolerable. In the case of osteoporotic and comminuted trochanteric bone fragments, non-absorbable suture procedures should also be used [25].

The challenges faced in reconstructing the large bone loss in the proximal femur during surgery are one of the most prominent problems in treating IFFs with primary arthroplasty. Cemented endoprostheses can be beneficial in patients with a thin cortical bone structure to reduce the risk of periprosthetic fractures. Cement application, on the other hand, may increase cardiopulmonary risks in such patients [26,27]. Aside from the bone loss, there are studies suggesting the use of larger femoral stems with long calcar support, due to the wide femoral canal observed in older adult patients [26,28,29]. Cementless long femoral stems, which can be applied as monoblock and modular options with the principle of diaphyseal engagement, can be shaped in a cylindrical or tapered form. The success of tapered porous covered femoral stems, in terms of their resistance to implant failure and bone on-growth, has been demonstrated [30]. On the other hand, rectangular short stem endoprostheses have been stated to yield satisfactory results in IFF with metaphyseal engagement [31].

In their prospective study, Bonneviallea et al. [32] found that, apart from increased bleeding, there was no significant difference in perioperative mortality and overall complication rates between primary arthroplasty and proximal femoral nail in patients with unstable IFF over 75 years of age [32]. Despite higher mortality rates in the first year, rates of implant-related complications and reoperations were shown, in another study, to be significantly lower in treatment of IFF with primary arthroplasty than internal fixation techniques [18]. Essentially, reoperation rates following internal fixation of IFF is reported between 0.5% to 56% depending on the age, fracture type and fixation device [33,34,35]. We believe that the experience acquired by surgeons with ten years or more of clinical practice may influence their arthroplasty attitudes in patients over 70 years of age, particularly in patients with increased fracture instability, in terms of lowering reoperation rates.

Preoperative comorbidities are also critical for patients that underwent an operation due to hip fractures to achieve the optimum functional recovery level following surgery, in addition to surgical techniques. In this sense, ASA scoring has a high predictive value for morbidity, mortality, and complications that may occur in patients during the perioperative phase [36]. Capkin et al. [37] concluded that an ASA score ≥3 is related to mortality. In our study, it is seen that as the ASA score rises, physicians’ preferences toward arthroplasty become more pronounced.

Despite the high response rates and consistent findings that we received from respondents, our survey study has some limitations. A responder bias may have occurred when participants marked different answer choices for a question they did not want to answer. In addition, the numbers of years in the profession may not always indicate the surgeon’s experience in hip trauma surgery. There is no consensus on which classification system is superior to define stable or unstable intertrochanteric fractures; thus, we did not use any classification system or representative fracture drawings in the scenarios [13]. Demonstrative hip X-rays were chosen based on the numbers of fracture fragments to define instability. However, differences in treatment attitudes may have occurred from the fact that bone quality was not specified visually or textually in the patient scenarios. Due to the small number of physicians who took part in our questionnaire, the findings could not be representative of all orthopedics and traumatology specialists in our country.

## 5. Conclusions

The treatment attitudes towards IFF of orthopedic specialists and specialty students in our country were examined in this questionnaire study, which did not aim to provide guidance on which treatment option leads to better results. In our sample of Turkish surgeons, the primary arthroplasty was considered a valuable alternative approach for unstable intertrochanteric fractures in older adults, especially in those who need early mobilization and have high ASA scores.

## Figures and Tables

**Figure 1 geriatrics-07-00018-f001:**
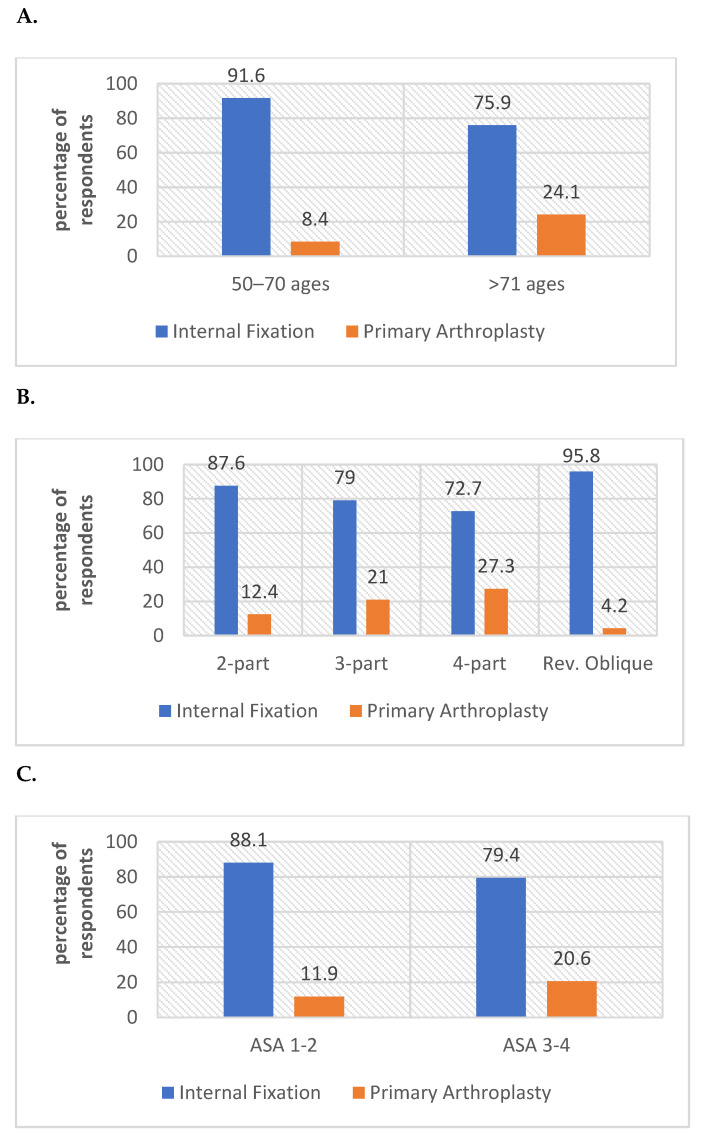
Surgical attitudes of respondents based on age (**A**), IFF type (**B**) and ASA score (**C**).

**Table 1 geriatrics-07-00018-t001:** Demonstrative X-rays and patient scenarios.

Demonstrative X-rays	Patient Scenarios
**For X-ray 1, four scenarios were developed.** 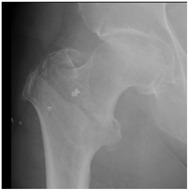	**Scenario 1**: What would be your operation preference for a mobile patient aged 50–70 and with an ASA score of 1–2? **Scenario 2**: What would be your operation preference for a mobile patient aged 71 years or older with an ASA score of 1–2? **Scenario 3**: What would be your operation preference for a mobile patient aged 50–70 with an ASA score of 3–4? **Scenario 4**: What would be your operation preference for a mobile patient aged 71 years or older with an ASA score of 3–4?
**For X-ray 2, four scenarios were developed.** 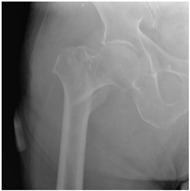	**Scenario 1**: What would be your operation preference for a mobile patient aged 50–70 and with an ASA score of 1–2? **Scenario 2**: What would be your operation preference for a mobile patient aged 71 years or older with an ASA score of 1–2? **Scenario 3**: What would be your operation preference for a mobile patient aged 50–70 with an ASA score of 3–4? **Scenario 4**: What would be your operation preference for a mobile patient aged 71 years or older with an ASA score of 3–4?
**For X-ray 3, four scenarios were developed.** 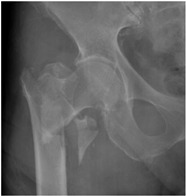	**Scenario 1**: What would be your operation preference for a mobile patient aged 50–70 and with an ASA score of 1–2? **Scenario 2**: What would be your operation preference for a mobile patient aged 71 years or older with an ASA score of 1–2? **Scenario 3**: What would be your operation preference for a mobile patient aged 50–70 with an ASA score of 3–4? **Scenario 4**: What would be your operation preference for a mobile patient aged 71 years or older with an ASA score of 3–4?
**For X-ray 4, four scenarios were developed.** 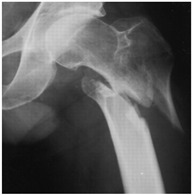	**Scenario 1**: What would be your operation preference for a mobile patient aged 50–70 and with an ASA score of 1–2? **Scenario 2**: What would be your operation preference for a mobile patient aged 71 years or older with an ASA score of 1–2? **Scenario 3**: What would be your operation preference for a mobile patient aged 50–70 with an ASA score of 3–4? **Scenario 4**: What would be your operation preference for a mobile patient aged 71 years or older with an ASA score of 3–4?

**Table 2 geriatrics-07-00018-t002:** Demographics of the respondents.

Survey Questions	Choices	*n*	%
**Current Practice Setting**	Training and Research Hospital	51	32
	University Hospital	28	18
	State Hospital	51	32
	Private Hospital	23	15
	Other	3	2
**Residency Training Institute**	Training and Research Hospital	44	28
	University Hospital	109	69
**Average Years in Practice**	1 to 10 years	101	64
	More than 10 years	55	34

**Table 3 geriatrics-07-00018-t003:** Comparison of surgical attitudes between respondents with academic and non-academic practice settings.

Scenario	Current Practice Setting	Internal Fixation	Arthroplasty	X^2^	Cramer’s V	*p*
**2-part** **ASA 1–2** **50–70 years**	Academic	79 (100.0%)	0 (0.0%)	1.033		0.494
	Non-academic	76 (98.7%)	1 (1.3%)		0.08	
**2-part** **ASA 1–2** **>70 years**	Academic	71 (89.9%)	8 (10.1%)	1.528		0.216
	Non-academic	64 (%83.1%)	13 (16.9%)		0.099	
**2-part** **ASA 3–4** **50–70 years**	Academic	73 (92.4%)	6 (7.6%)	0.752		0.386
	Non-academic	68 (88.3%)	9 (11.7%)		0.04	
**2-part** **ASA 3–4** **>70 years**	Academic	62 (78.5%)	17 (21.5%)	1.426		0.232
	Non-academic	54 (70.1%)	23 (29.9%)		0.09	
**3-part** **ASA 1–2** **50–70 years**	Academic	75 (94.9%)	4 (5.1%)	3.089		0.097
	Non-academic	66 (86.8%)	10 (13.2%)		0.13	
**3-part** **ASA 1–2** **>70 years**	Academic	64 (81.0%)	15 (19.0%)	3.086		0.079
	Non-academic	53 (68.8%)	24 (31.2%)		0.15	
**3-part** **ASA 3–4** **50–70 years**	Academic	68 (86.1%)	11 (13.9%)	0.086		0.770
	Non-academic	65 (84.4%)	12 (15.6%)		0.02	
**3-part** **ASA 3–4** **>70 years**	Academic	56 (70.9%)	23 (29.1%)	2.646		0.132
	Non-academic	45 (58.4%)	32 (41.6%)		0.13	
**4-part** **ASA 1–2** **50–70 years**	Academic	-	8 (10.1%)	0.313		0.576
	Non-academic	67 (87.0%)	10 (13.0%)		0.04	
**4-part** **ASA 1–2** **>70 years**	Academic	54 (69.2%)	24 (30.8%)	0.003		0.957
	Non-academic	53 (68.8%)	24 (31.2%)		0.008	
**4-part** **ASA 3–4** **50–70 years**	Academic	66 (83.5%)	13 (16.5%)	1.173	0.08	0.319
	Non-academic	59 (76.6%)	18 (23.4%)			
**4-part** **ASA 3–4** **>70 years**	Academic	42 (53.2%)	37 (46.8%)	0.000		0.992
	Non-academic	41 (53.2%)	36 (46.8%)		-	
**Rev. Oblique** **ASA 1–2** **50–70 years**	Academic	79 (100.0%)	-	-		-
	Non-academic	77 (100.0%)	-		-	
**Rev. Oblique** **ASA 1–2** **>70 years**	Academic	77 (97.5%)	2 (2.5%)	1.428		0.232
	Non-academic	72 (93.5%)	5 (6.5%)		0.09	
**Rev. Oblique** **ASA 3–4** **50–70 years**	Academic	78 (100.0%)	0 (0.0%)	2.052		0.245
	Non-academic	75 (97.4%)	2 (2.6%)		0.11	
**Rev. Oblique** **ASA 3–4** **>70 years**	Academic	71 (91.0%)	7 (9.0%)	0.639		0.424
	Non-academic	67(89.0%)	10(13.0%)		0.06	

**Table 4 geriatrics-07-00018-t004:** Comparison of surgical attitudes between respondents with 1 to 10 years and more than 10 years of experience in practice.

Scenario	Average Number of Years in Practice	Internal Fixation	Arthroplasty	X^2^	Cramer’s V	*p*
**2-part** **ASA 1–2** **50–70 years**	1–10 years	101 (100.0%)	0 (0.0%)	1.783		0.361 ^a^
	>10 years	56 (98.2%)	1 (1.8%)		0.10	
**2-part** **ASA 1–2** **>71 years**	1–10 years	93 (92.1%)	8 (7.9%)	7.006		0.008 *
	>10 years	44 (77.2%)	13 (22.8%)		0.21	
**2-part** **ASA 3–4** **50–70 years**	1–10 years	91 (90.1%)	10 (9.9%)	0.215		0.643
	>10 years	50 (87.7%)	7 (12.3%)		0.03	
**2-part** **ASA 3–4** **>71 years**	1–10 years	79 (78.2%)	22 (21.8%)	3.305		0.069
	>10 years	37 (64.9%)	20 (35.1%)		0.15	
**3-part** **ASA 1–2** **50–70 years**	1–10 years	92 (92.0%)	8 (8.0%)	1.445		0.229
	>10 years	49 (86.0%)	8 (14.0%)		0.10	
**3-part** **ASA 1–2** **>71 years**	1–10 years	81 (80.2%)	20 (19.8%)	5.506		0.019 *
	>10 years	36 (63.2%)	21 (36.8%)		0.21	
**3-part** **ASA 3–4** **50–70 years**	1–10 years	91 (90.1%)	10 (9.9%)	4.880		0.027 *
	>10 years	44 (77.2%)	13 (22.8%)		0.18	
**3-part** **ASA 3–4** **>71 years**	1–10 years	70 (69.3%)	31 (30.7%)	2.091		0.148
	>10 years	33 (57.9%)	24 (42.1%)		0.12	
**4-part** **ASA 1–2** **50–70 years**	1–10 years	92 (91.1%)	9 (8.9%)	1.708		0.191
	>10 years	48 (84.2%)	9 (15.8%)		0.10	
**4-part** **ASA 1–2** **>71 years**	1–10 years	74 (74.0%)	26 (26.0%)	4.339		0.037 *
	>10 years	33 (57.9%)	24 (42.1%)		0.17	
**4-part** **ASA 3–4** **50–70 years**	1–10 years	84 (83.2%)	17 (16.8%)	1.380		0.240
	>10 years	43 (75.4%)	14 (24.6%)		0.10	
**4-part** **ASA 3–4** **>71 years**	1–10 years	60 (59.4%)	41 (40.6%)	5.306		0.021 *
	>10 years	23 (40.4%)	34 (59.6%)		0.19	
**Rev. Oblique** **ASA 1–2** **50–70 years**	1–10 years	101 (100.0%)	-	-		-
	>10 years	57 (100.0%)	-		-	
**Rev. Oblique** **ASA 1–2** **>71 years**	1–10 years	97 (96.0%)	4 (4.0%)	1.570		0.285 ^a^
	>10 years	52 (91.2%)	5 (8.8%)		0.10	
**Rev. Oblique** **ASA 3–4** **50–70 years**	1–10 years	97 (97.0%)	3 (3.0%)	0.227		1.000 ^a^
	>10 years	56 (98.2%)	1 (1.8%)		0.03	
**Rev. Oblique** **ASA 3–4** **>71 years**	1–10 years	92 (92.0%)	8 (8.0%)	2.281		0.131
	>10 years	48 (84.2%)	9 (15.8%)		0.12	

^a^: Fisher’s exact test, * *p* < 0.05.

## Data Availability

Not applicable.

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
