# Peer review of "Primary Arthroplasty or Internal Fixation in Intertrochanteric Femur Fractures: A Survey of Surgical Attitudes of Orthopedic Surgeons in Turkey"

_geriatrics, 2022, doi:10.3390/geriatrics7010018_

Round 1
Reviewer 1 Report
It is a study that assesses whether orthopedic surgeons in Turkey treat pertrochanteric fractures with osteosynthesis or arthroplasty. They survey 159 surgeons online. They basically evaluate the experience of the surgery, type of fracture (2-, 3-, 4-parts, reverse oblique fracture), ASA (1-2 vs 3-4), age of the patient (50-70 vs> 70). They conclude that arthroplasty is an option for the treatment of pertrochanteric fractures, especially in patients> 71 years of age, ASA 3-4, unstable fractures and that this is indicated more by experienced surgeons (> 10 years) than those of 1 -10 years. The% of surgeons who report arthroplasty are 12.4, 21, 27.3 and 4.2, in 2-, 3-, 4-part fractures, reverse oblique fracture, respectively
The truth is that it is an amazing study from my point of view. I work in a hospital where we operate on about 500 pertrochanteric fractures a year, and in the 4 types of fractures studied in this article, all of them would be treated with osteosynthesis, and none of them with arthroplasty. So for me, the result is amazing.
1.- Title: Add Turkey to title.
2.- Throughout the text there are different places where there is extra space:Results:… 38.2 (SD: 7.9). Thirty-two percent (n: 51).Talk:… 239 German orthopedic specialists, but primary arthroplasty was not pro-vided as a treatment choice to the participants in the study. Although the participant’s additional treatment recommendations were asked in a separate field….… Demonstrative hip x-rays were chosen based on the numbers of fracture fragments to de-fine instability. However, differences in treatment attitudes may have occurred from the fact that bone quality was not specified visually or textually in the patient scenarios. Because of the small number of physicians….
3.- Discussion:Bonneviallea et al. found that,Add the reference
4.- The conclusion must be changed, the conclusion must respond to the stated objective which is this:The aim of this survey study is to examine the surgical method preferences of Turkish orthopedics and traumatology specialists and residents in patients with IFF of various ages, comorbidity situations and fracture types, using an internet-based questionnaire and to assess the findings in light of current literature.Current Conclusion: In light of our present findings, primary arthroplasty can be considered a valuable alternative approach for Turkish surgeons.This is not the conclusion; the conclusion is to put the results they have obtained. Prosthesis is not an alternative, prosthesis is used in% of patients.
Personally, I disagree with the results of the study, but the results of the survey are what they are.
Author Response
1. “Turkey” is added to the title
2. The extra spaces were erased and corrected.
3. The reference number was added.
4. The conclusion section was changed.
Reviewer 2 Report
It is an interesant study. The paper is well structured. The results are clear. May be it is necessary to introduce in discusitions a short topic af the surgical technique and aproach (minim-invazive or not).
Author Response
Thank you for evaluating the article and for your constructive criticism.
- Since the study was a questionnaire study and the questions were answered by different surgeons, a single surgical technique was not used. Since such a question was not included in the questionnaire, the surgical technique was not mentioned in the discussion section.
Reviewer 3 Report
The paper “Primary Arthroplasty in Intertrochanteric Femur Fractures: A Survey of Surgical Attitudes of Orthopedic Surgeons” deals with interesting problem. However I have some comments.
- First, it would suggest to change the title – the paper is about preferences of orthopedic surgeons, not primary arthroplasty itself.
- Have you got any additional information about the sex structure of the group under the study, position and specialization of respondents? In discussion you have indicated “high response rate” but “small number of physicians who took part in this survey” - so How have you assessed the response rate? How this sample “fits to” general population of orthopedic surgeons.
- Add the information which kind of professional experience you have in mind in this study.
- There is missing paragraph about the statistical methods used in this analysis. Add the information which statistical software was used to perform analysis.
- The description of the results should be improved – for example
- “arthroplasty preferences of respondents with 10 years or more of professional experience were observed to be statistically significant than the respondents with 1-10 years of experience” - What does it mean that preferences were statistically significant?
- “were more likely to have arthroplasty attitudes” – what do you have in mind saying arthroplasty attitude?
- “preference based on age” – it should be clearly indicated that you have in mind patient’s age.
- Add the information about the missing values in the data presented in tables – for example for “Residency Training Institute” n=153, length of practice N=156, current practice setting =156, some missing data in table 2, in table 4 you have indicated 57 respondents in >10 years category, but in table 2 you have declared only 55, etc.
- Tables – “%” symbol should be displayed after the number, not before
- Table 4 – “parcali” should be translated to English
- Figure 1 – What kind of data have been presented there? One should have in mind that if the answers were “summed up” for all scenarios in respect to age, etc. multiple answers from one respondent are summed up – the individual knowledge, experience, etc. of the respondents can influence the results as well.
- The conclusion about arthroplasty as a valuable alternative approach for Turkish surgeons is not based on the presented results. You have analyzed preferences (declarative) of respondents not validity of this approach.
- Some references are very old – maybe you can find out any newer ones.
- In abstract the ASA has been used without explanation while used for the first time.
Author Response
- The title was changed.
- Current practice setting was evaluated as academic or non-academic. The sex structure of the group was not evaluated. By high response rates, it is meant that the majority of questions are answered by the participants, and the expression "small number of physicians who took part in this survey" was used since the survey participation rate was approximately 10% (159/1553) of the orthopedists registered in the system. This was one of the limitation of this study that was mentioned in the discussion section.
- Years of experience was mentioned.
- Statistical methods and software used in the analysis were added into Material and methods section.
- Preferences of respondents with 10 years or more of professional experience were found to have a higher rate of arthroplasty attitudes than respondents working 1 to 10 years of experience. The difference is statistically significant.
- Arthroplasty preferences
- We have mentioned patient’s age with “prefernce based on age”. It was added into text and highlighted.
- Since some participants did not answer all of the questions, the number of participants in some questions seems different and is presented in this way in the table.
- “%” symbol was displayed after after the number in tables.
- Table 4- “parcali” was translated to English.
- Surgical attitudes of respondents based on age (A), IFF type (B) and ASA score (C) were compared between internal fixation and arthroplasty.
- The conclusion section was changed.
- Updated and new references were added.
- ASA is clearly stated in abstract section. It is written with its explanation.